# Clinical Significance of a 16S-rDNA Analysis of Heart Valves in Patients with Infective Endocarditis: a Retrospective Study

Gustav Johansson,[a] Torgny Sunnerhagen,[a,b,c] Sigurdur Ragnarsson,[d,e] Magnus Rasmussen[a,f]

[a]Division of Infection Medicine, Department of Clinical Sciences Lund, Lund University, Lund, Sweden

[b]Department of Clinical Microbiology, Infection Control and Prevention, Office for Medial Services, Region Skåne, Lund, Sweden

[c]Department of Clinical Microbiology, Copenhagen University Hospital, Rigshospitalet, Copenhagen, Denmark

[d]Division of Cardiothoracic Surgery, Department of Clinical Sciences Lund, Lund University, Lund, Sweden

[e]Department of Cardiothoracic and Vascular Surgery, Skåne University Hospital, Lund, Sweden

[f]Department of Infectious Diseases, Skåne University Hospital, Lund, Sweden

**ABSTRACT** A substantial proportion of patients with infective endocarditis (IE) are subjected to heart valve surgery. Microbiological findings on valves are important both for diagnostics and for tailored antibiotic therapy, post-operatively. The aims of this study were to describe microbiological findings on surgically removed valves and to examine the diagnostic benefits of 16S-rDNA PCR and sequencing (16S-analysis). Adult patients who were subjected to heart valve surgery for IE between 2012 and 2021 at Skåne University Hospital, Lund, where a 16S-analysis had been performed on the valve, constituted the study population. Data were gathered from medical records, and the results from blood cultures, valve cultures, and 16S-analyses of valves were compared. A diagnostic benefit was defined as providing an agent in blood culture negative endocarditis, providing a new agent in episodes with positive blood cultures, or confirming one of the findings in episodes with a discrepancy between blood and valve cultures. 279 episodes in 272 patients were included in the final analysis. Blood cultures were positive in 259 episodes (94%), valve cultures in 60 episodes (22%), and 16S-analyses in 227 episodes (81%). Concordance between the blood cultures and the 16S-analysis was found in 214 episodes (77%). The 16S-analyses provided a diagnostic benefit in 25 (9.0%) of the episodes. In blood culture negative endocarditis, the 16S-analyses had a diagnostic benefit in 15 (75%) of the episodes. A 16S-analysis should be routinely performed on surgically removed valves in blood culture negative endocarditis. In patients with positive blood cultures, 16S-analysis may also be considered, as a diagnostic benefit was provided in some patients.

**IMPORTANCE** This work demonstrates that it can be of importance to perform both cultures and analysis using 16S-rDNA PCR and sequencing of valves excised from patients undergoing surgery for infective endocarditis. 16S-analysis may help both to establish a microbiological etiology in cases of blood culture negative endocarditis and to provide help in situations where there are discrepancies between valve and blood cultures. In addition, our results show a high degree of concordance between blood cultures and 16S-analyses, indicating that the latter has a high sensitivity and specificity for the etiological diagnosis of endocarditis in patients who were subjected to heart valve surgery.

**KEYWORDS** diagnostic benefit, 16S-rDNA analysis, infective endocarditis, 16S RNA, species determination, surgery

Infective endocarditis (IE) is a severe infection of the heart valves with an in-hospital mortality rate that is slightly higher than 10% in high-income countries, such as Sweden (1, 2). The most important diagnostic tools in IE are blood cultures and echocardiography (3). Valve surgery is necessary in a notable proportion of patients with IE, varying from 13% to 52% of the episodes in different reports (2, 4). Further etiological diagnostics can be

Address correspondence to Magnus Rasmussen, magnus.rasmussen@med.lu.se.

The authors declare no conflict of interest.

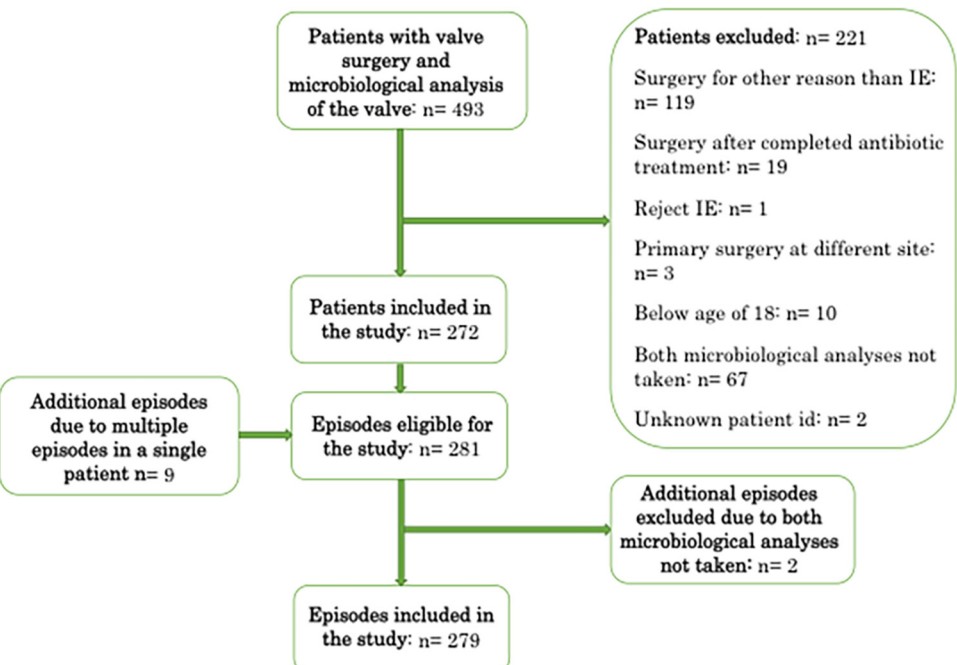

**FIG 1** Flowchart illustrating the development of the study population.

made through valve cultures as well as valve 16S-rDNA PCR and sequencing (16S-analysis) in patients who undergo surgery (5). Microbiological findings are of importance in the choice of antibiotic treatment, which should be tailored to the causative agent. Valve cultures are also used to determine the duration of postoperative antibiotic treatment, with positive valve cultures indicating a need for a new, full course treatment (3).

Blood culture negative endocarditis (BCNE) constitutes 9 to 24% of all episodes of endocarditis (6–8). Negative blood cultures can be the result of a previous antibiotic treatment or infection caused by fastidious or intracellular bacteria, such as *Tropheryma whipplei*, *Coxiella burnetii*, or *Bartonella* spp. (6). Serological analyses and 16S-analyses on surgically removed valves are alternative methods of finding the causative agents in these patients.

The aim of this study was to describe the microbiological findings in IE patients who were subjected to surgery at our center and evaluate the potential additional value of performing 16S-analyses for the management of IE.

## RESULTS

**Study population and patient characteristics.** In total, 493 patients were eligible for the study. 221 patients were excluded, and the reasons for the exclusions are illustrated in Fig. 1. The episodes that were excluded due to not having both microbiological analyses performed are shown, along with the available microbiological findings, in Appendix 1. Finally, 272 patients with 279 episodes of IE were included in the study. Three patients had two episodes of IE, and two patients had three episodes each. The patient characteristics for all of the episodes are presented in Table 1.

**Findings in microbiological analyses.** Blood cultures were performed in all but three of the episodes, whereas valve cultures and 16S-analyses were performed in all of the episodes. A total of seven valve cultures and two 16S-analyses were defined as contaminated, as presented in full in Appendix 2. A full presentation of the microbiological findings is given in Table 2 and Appendix 3. Blood cultures detected a possible causative agent in 259 episodes (94%), valve cultures in 60 episodes (22%), and 16S-analyses in 227 episodes (81%). The most common agent found in blood cultures was viridans streptococci (*n* = 71), which was followed by *S. aureus* (*n* = 70). The most common agent found in valve cultures was *S. aureus* (*n* = 25), which was followed by enterococci (*n* = 8) and coagulase-negative staphylococci (CoNS) (*n* = 8). The most common agent found in the 16S-analyses was *S. aureus* (*n* = 66), which was followed by viridans streptococci (*n* = 57) and enterococci

**TABLE 1** Patient characteristics in the whole study population[a]

| Patient characteristic | Total (279) median (IQR) or n (%) |
|---|---|
| Definite IE | 243 (87) |
| Age (yrs) | 66 (52 to 74) |
| Sex (male) | 201 (72) |
| Immunosuppression | 22 (7.9) |
| | |
| Charlson score | |
| 0 to 1 | 213 (76) |
| 2 to 4 | 59 (21) |
| 5 to 9 | 7 (2.5) |
| Predisposing factors | 151 (54) |
| Prosthetic valve | 80 (29) |
| Previous IE | 25 (9.0) |
| Valvular heart disease | 81 (29) |
| Intravenous drug use | 11 (3.9) |
| | |
| Affected Valve | |
| Aortic valve | 147 (53) |
| Mitral valve | 80 (29) |
| Pulmonary valve | 8 (2.9) |
| Tricuspid valve | 4 (1.4) |
| Aortic & Mitral valve | 33 (11) |
| Aortic & Tricuspid valve | 5 (1.5) |
| Mitral & Tricuspid valve | 1 (1.2) |
| Aortic & Pulmonary valve | 1 (0.3) |

[a]IQR, interquartile range; IE, infective endocarditis.

($n$ = 20). The proportion of viridans streptococci among the positive analyses was significantly higher in the blood cultures (28%) and 16S-analyses (25%), compared to the valve cultures (6%) ($P$ = 0.001 and $P$ = 0.004 for the differences, respectively, using Fisher's exact test). HACEK-group agents were also more commonly present in the positive blood cultures and 16S-analyses than in the valve cultures. However, this difference was not statistically significant.

A large overlap of positive analyses (Fig. 2), irrespective of agent, was seen between the three analyses. Most episodes with an etiological diagnosis had a positive blood culture,

**TABLE 2** Findings in microbiological analyses[a]

| Finding | Blood cultures ($n$ = 276), n (%) | Valve cultures ($n$ = 279), n (%) | 16S-analysis ($n$ = 279), n (%) |
|---|---|---|---|
| Positive result | 259 (94) | 60 (22) | 227 (81) |
| | | | |
| Causative agents | | | |
| *S. aureus* | 70 (27) | 25 (42) | 66 (29) |
| Viridans Streptococci | 71 (27) | 5 (8.3) | 57 (25) |
| Enterococci | 25 (9.7) | 8 (13) | 20 (8.8) |
| CoNS | 25 (9.7) | 8 (13) | 16 (7.0) |
| HACEK-group | 10 (3.9) | 0 (0) | 10 (4.4) |
| *S. bovis* | 13 (5.0) | 2 (3.3) | 11 (4.8) |
| $\beta$-hemolytic streptococci | 17 (6.6) | 3 (5.0) | 19 (8.4) |
| Others | 26 (10)[b] | 5 (8.3)[c] | 25 (11)[d] |
| Polymicrobial | 2 (0.8) | 4 (6.7) | 3 (1.3) |

[a]Causative agents are presented as percentages of positive results. CoNS, coagulase negative staphylococci. The species of the viridans streptococci, enterococci, CoNS, HACEK-group, *S. bovis*, and $\beta$-hemolytic streptococci are shown in Appendix 3.
[b]The other agents that were found in the blood cultures were *Abiotrophia defectiva* ($n$ = 2), *Aerococcus urinae* ($n$ = 4), *Candida albicans* ($n$ = 1), *Capnocytophaga canimorsus* ($n$ = 1), *Corynebacterium* spp. ($n$ = 4), *Cutibacterium acnes* ($n$ = 1), *Enterobacter cloacae* ($n$ = 1), *Escherichia coli* ($n$ = 1), *Granulicatella* spp. ($n$ = 2), *Rothia mucilaginosa* ($n$ = 2), *Streptococcus pneumoniae* ($n$ = 5), *Shewanella algae* ($n$ = 1), and *Veillonella parvula* ($n$ = 1).
[c]The other agents that were found in the valve cultures consisted of *C. albicans* ($n$ = 1), *C. acnes* ($n$ = 1), *E. cloacae* ($n$ = 1), *E. coli* ($n$ = 1), and *Pantoea* spp. ($n$ = 1).
[d]The other agents that were found in the 16S-analyses were *A. defectiva* ($n$ = 2), *A. urinae* ($n$ = 4), *C. canimorsus* ($n$ = 1), *Corynebacterium* spp. ($n$ = 4), *C. acnes* ($n$ = 1), *E. cloacae* ($n$ = 1), *E. coli* ($n$ = 1), *Granulicatella* spp. ($n$ = 1), *Rothia dentocariosa* ($n$ = 1), *R. mucilaginosa* ($n$ = 2), *S. pneumoniae* ($n$ = 5) and *Tropheryma whipplei* ($n$ = 2).

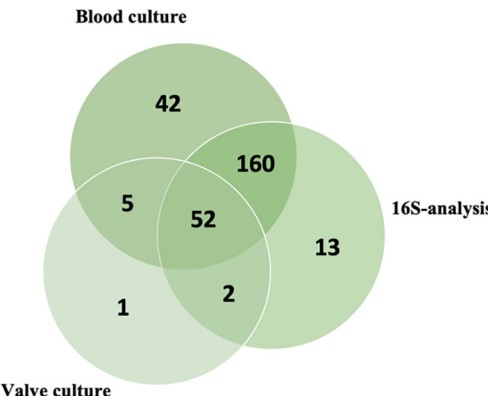

**FIG 2** Venn diagram showing the distribution of positive analyses. In four episodes, no analysis was positive.

a positive 16S-analysis, or a combination of the two. The valve culture was the only source of etiological importance in only one episode, whereas the blood culture and 16S-analysis were the only sources in 42 and 13 episodes, respectively. The blood cultures were negative in a total of 20 episodes (7.2%). A causative agent was found in 16 (80%) of the BCNE episodes, whereas in 4 episodes, all of the microbiological analyses were negative. Polymicrobial findings were more frequent in the valve cultures, with 6.7% positive analyses, compared to 0.8% in the blood cultures ($P = 0.01$ for the difference, using Fisher's exact test) and 1.3% in the 16S-analyses ($P = 0.04$ for the difference, using Fisher's exact test).

**Diagnostic benefit of a 16S-analysis.** As previously shown in Table 2, a 16S-analysis found an agent in 81% of the episodes. The 16S-analyses were concordant with the blood cultures in 214 out of 279 episodes (77%). In the whole study population, a diagnostic benefit was seen in 25 out of 279 episodes (9.0%). A diagnostic benefit was significantly more common in BCNE (15 out of 20 episodes, 75%), compared to episodes with positive blood cultures (10 out of 259 episodes, 3.9%) ($P < 0.001$). A full presentation of the diagnostic benefits of the 16S-analyses as well as the microbiological findings in those episodes are shown in Fig. 3. In BCNE with a diagnostic benefit, a variety of agents were found in the 16S-analyses. Most notably, there were two findings of *T. whipplei*. Another of the episodes in which a 16S-analysis provided a benefit was in a patient with BCNE who had had *S. bovis* IE three years prior, with findings of *S. bovis* in the 16S-analysis in the included episode.

In three episodes, a 16S-analysis detected a different possible agent in patients with positive blood cultures. In another two cases with positive blood cultures, a 16S-analysis demonstrated the same bacterial agent as in blood cultures, with the additional finding of *S. aureus*. In seven cases in which there was a discrepancy between the blood and valve cultures, a 16S-analysis confirmed one of the microbiological findings (Fig. 3).

## DISCUSSION

In this study, blood cultures, valve cultures, and 16S-analyses identified a causative agent in 94%, 22%, and 81% of episodes, respectively. We found a 16S-analysis to have a diagnostic benefit in 9.0% of all episodes of IE and in 75% of BCNE cases.

The proportion of positive blood cultures was higher, at 94% of the episodes, compared to the results of other studies, which reported a lower diagnostic yield of 76 to 87% (7, 9). Valve cultures and 16S-analyses had similar diagnostic yields, compared to the results of previous studies (22% and 81% respectively), compared to 21 to 26% and 68 to 87% (7–9). The distribution of causative agents found in the different microbiological analyses has a pattern that is similar to those observed in other studies, with *S. aureus* being detected in a large proportion of the valve cultures (7–10). In our study, viridans streptococci were rarely isolated via valve culture (6.0% of positive cultures), compared to the results of other studies, in which viridans streptococci constituted around 15% of the positive valve cultures (7, 10). The spectrum of causative agents found in blood

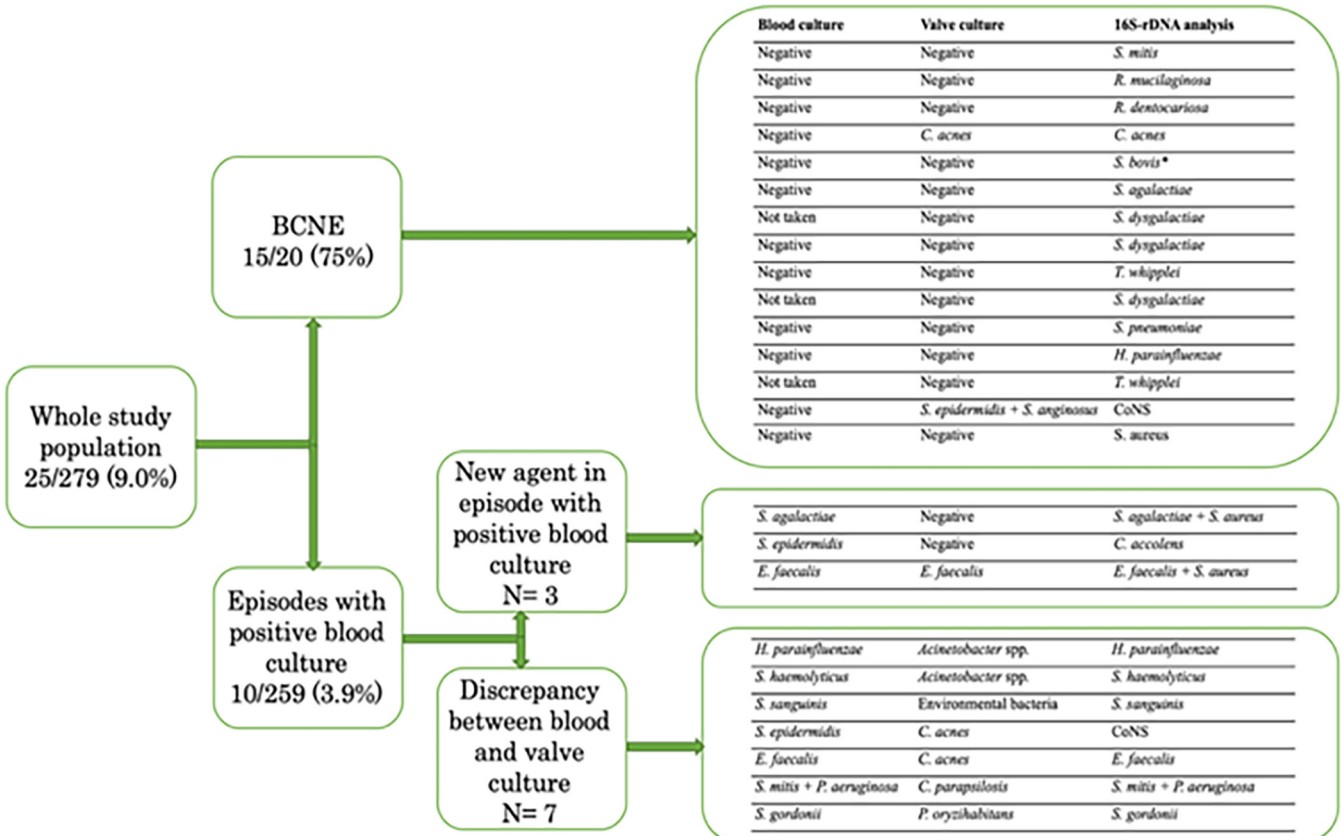

| Blood culture | Valve culture | 16S-rDNA analysis |
|---|---|---|
| Negative | Negative | S. mitis |
| Negative | Negative | R. mucilaginosa |
| Negative | Negative | R. dentocariosa |
| Negative | C. acnes | C. acnes |
| Negative | Negative | S. bovis* |
| Negative | Negative | S. agalactiae |
| Not taken | Negative | S. dysgalactiae |
| Negative | Negative | S. dysgalactiae |
| Negative | Negative | T. whipplei |
| Not taken | Negative | S. dysgalactiae |
| Negative | Negative | S. pneumoniae |
| Negative | Negative | H. parainfluenzae |
| Not taken | Negative | T. whipplei |
| Negative | S. epidermidis + S. anginosus | CoNS |
| Negative | Negative | S. aureus |

| | | |
|---|---|---|
| S. agalactiae | Negative | S. agalactiae + S. aureus |
| S. epidermidis | Negative | C. accolens |
| E. faecalis | E. faecalis | E. faecalis + S. aureus |

| | | |
|---|---|---|
| H. parainfluenzae | Acinetobacter spp. | H. parainfluenzae |
| S. haemolyticus | Acinetobacter spp. | S. haemolyticus |
| S. sanguinis | Environmental bacteria | S. sanguinis |
| S. epidermidis | C. acnes | CoNS |
| E. faecalis | C. acnes | E. faecalis |
| S. mitis + P. aeruginosa | C. parapsilosis | S. mitis + P. aeruginosa |
| S. gordonii | P. oryzihabitans | S. gordonii |

**FIG 3** Diagnostic benefits of 16S-analyses, divided into different types of benefits and with the findings of the microbiological analyses for every episode. An asterisk indicates that the patient was diagnosed with *S. bovis* IE three years prior to the included episode.

cultures and 16S-analyses seems to be relatively similar, which has also been observed in previous studies (7, 9).

With our definition of contamination of valve analyses, nine contaminations were found, with seven being in a valve culture and two being in a 16S-analysis. All analyses were defined as contaminated demonstrated agents in which contamination is not unreasonable. Some agents, such as environmental bacteria, are likely to represent contamination, whereas other agents, such as *Cutibacterium acnes* and *Candida parapsilosis*, can represent contamination but can, of course, also be agents that are responsible for IE. Early concerns that 16S-analyses are too sensitive (11) are not supported by our findings. Instead, we found that the risk of contamination was higher in valve cultures than in 16S-analyses. In particular, valve cultures that were deemed to be contaminated demonstrated the growth of water-living bacteria, such as *Pseudomonas oryzihabitans* and *Acinetobacter* species. Polymicrobial findings, possibly representing partial contamination, were more frequent in valve cultures, too. This is in accordance with the results of Voldstedlund et al. (12), who found valve cultures to be contaminated in 35% of episodes, whereas there were no contaminations in 16S-analyses. However, this should be interpreted while keeping the different definitions of contamination between studies in mind.

Two patients with *Candida* in cultures were included in this study. One of them, who had *C. parapsilosis* in a valve culture, was defined as contamination. The second patient had findings of *C. albicans* in a blood culture and a valve culture. It is important to note that it would be impossible for *Candida* to be positive in all three microbiological analyses. Data about ITS or 18S-rDNA analyses were not systematically collected, and the analyses were rarely performed.

Findings of bacterial DNA on valves have previously been reported to persist for long periods of time after IE, despite a successful antibiotic treatment (13, 14). In one of our cases with a diagnostic benefit, *S. bovis* was found in a 16S-analysis. This patient had

previously been diagnosed with IE that had been caused by *S. bovis* three years prior to the included episode. By our definition, this patient was classified as having a diagnostic benefit from a 16S-analysis. In practice, the finding of *S. bovis* DNA could also be explained by the previous episode of IE, meaning that a 16S-analysis must be interpreted with caution and in relation to the clinical presentation.

16S-analyses showed concordance with blood cultures in 77% of the episodes. Armstrong et al. (8) found a lower concordance: 43% of the episodes. Their study included a larger portion of BCNE, which, by their definitions, could not be concordant. Our findings with a high concordance between blood cultures and 16S-analyses in individual episodes, in combination with the previously discussed finding of the similar spectrum of microbiological findings, suggest that the 16S-analysis is a reliable diagnostic tool to find agents in IE.

In our study, a 16S-analysis had a diagnostic benefit in only 9.0% of all episodes of IE. In contrast to our findings, previous studies have presented data of a diagnostic benefit in 13 to 31% of the episodes (7, 8, 15). The differences between the results of Armstrong et al. (8), Rodríguez-García et al. (15), and our study could possibly be explained by different definitions of diagnostic benefits. They considered the confirmation by a 16S-analysis of a previously found skin commensal to be of diagnostic benefit. Halavaara et al. (7), like us, considered episodes where a 16S-analysis confirmed a causative agent found in a blood culture or a valve culture when a discrepancy was found between the two to be of diagnostic benefit. The lower number of BCNE in our study could also explain some of the difference, as most of the benefits of 16S-analyses are seen in BCNE.

In episodes with BCNE, 16S-detected a possible causative agent in 75% of the episodes, similar to what previous studies have suggested (52 to 77%) (5, 7, 8). The findings from our study suggest limited diagnostic benefits of 16S-analysess in patients who already have positive blood cultures. However, a substantial benefit was seen in BCNE.

The importance of 16S-analysis in finding fastidious and intracellular microorganisms have been presented in multiple previous studies (6, 8, 15). The prevalence of BCNE caused by *Bartonella* spp. and *C. burnetii* is high in Southern Europe (6, 15), Asia (16, 17), and Northern Africa (18, 19). In contrast, findings of these agents in BCNE are extremely rare in the Scandinavian countries (12, 20, 21). Only one episode of IE that was caused by *C. burnetii* (20) and one episode of IE that was caused by *Bartonella* spp. (22) have been reported in Sweden, and we found no cases of either in our cohort. Two episodes of *T. whipplei* IE were diagnosed through 16S-analyses and, for these patients, the treatment was much affected by this microbiological finding. These findings highlight the importance of 16S-analyses, even in populations with a low prevalence of *Bartonella* spp. and *C. burnetii*, such as those of the Scandinavian countries. The absence of *Bartonella* spp. and *C. burnetii* also impacts the diagnostic benefits. Studies in which the mentioned bacteria are prevalent report a larger proportion of diagnostic benefits (7, 8, 15).

A strength of the study is that it is, to our knowledge, the largest study examining the microbiological findings on valves and the diagnostic benefits of 16S-analyses. Moreover, it is population-based, as there is just one cardiac surgery department that is serving a defined population. Since 16S-analyses were not routinely performed during the whole study period, the study population is most likely subject to a selection bias, where the patients with higher probabilities of diagnostic benefits would be selected for the analysis. The patients who were excluded due to missing microbiological analyses can be seen in Appendix 1. Since the study spanned a 10-year period, the protocol for microbiological analyses, diagnostics, and treatment could differ in between episodes.

**Conclusions.** A 16S-analysis should be routinely performed on surgically removed valves in BCNE. In patients with positive blood cultures, a 16S-analysis of valves could also be considered, as a diagnostic benefit was provided in a subset of patients.

## MATERIALS AND METHODS

**Study population.** Patients were retrieved through a search of all of the microbiological analyses that were performed by the Clinical Microbiology Laboratory at Region Skåne Medical Services, Lund, on specimens from the Department of Cardiothoracic Surgery at Skåne University Hospital (SUS) in Lund, Sweden, between

2012 and 2021. To be eligible for the study, the patient had to have undergone valve surgery and the microbiological analysis of the valve. This study was approved by the Swedish Ethical Review Authority (2021-05713), which waived the need for informed consent.

The exclusion criteria were as follows. (i) Patients who were subjected to surgery without suspicion of IE. (ii) Patients who underwent surgery due to IE after completing their antibiotic treatment. (iii) Episodes of suspected IE that were defined as reject IE, according to the ESC 2015 modified criteria (3), before surgery. (iv) Patients who did not undergo primary surgery at SUS. (v) Patients below the age of 18. (vi) Patients for whom either the 16S-analysis or the valve culture was missing.

**Definitions.** Medical records were retrospectively reviewed with a protocol of predetermined variables, including patient demographic data, comorbidities, data on presentation, treatment, work-up, and outcomes. Episodes were defined as definite, possible, or reject IE by using the ESC 2015 modified criteria (3). Data were gathered about blood cultures, valve cultures, and 16S-analyses. A microbiological finding on the valves was defined as contamination if (i) an ESC 2015 major criteria agent (*Staphylococcus aureus*, viridans streptococci, *Streptococcus bovis*, HACEK group, or enterococci) was found in the blood cultures with a different non-major agent in one valve analysis or (ii) if concordant findings of a causative agent in the blood cultures and the valve analysis (culture or 16S) did not match the findings of a positive finding in the other valve analysis. Concordance in a 16S-analysis was defined as having the same result in the blood cultures as in the 16S-analysis. A 16S-analysis was deemed to provide a diagnostic benefit if it demonstrated an agent in BCNE, if it found a new agent in episodes with positive blood cultures, or if there was a discrepancy between the blood and valve cultures and the 16S-analysis confirmed one of the findings.

**Microbiological analysis of heart valves.** At SUS, the sampling of heart valves for 16S-analyses and valve cultures has been routinely performed in patients who are undergoing valve surgery due to a suspicion of IE since 2019. Prior to that, valve cultures were performed routinely. 16S-analyses was performed frequently but not routinely. All valve cultures and 16S-analyses had previously been conducted as part of the clinical routine at the Clinical Microbiology Laboratory at Region Skåne Medical Services, Lund.

Valve tissue was divided in the surgical theater, using a pair of scissors under sterile conditions, into two equally large pieces. One piece was put into fastidious anaerobe broth (Neogen, NCM0199A) for culture, and the other piece was put into a dry sterile tube for a 16S-analysis. For culturing, the valve tissue was subjected to sonication for 90 s. Then, the broth was inoculated onto two agar plates for incubation in $CO_2$ enriched air (blood agar plate [Neogen, NCM2014B] with the addition of 40 mL of horse blood per liter of agar), in GCD (GC agar base [Difco, 228950] with the addition of 100 mL of horse blood and 40 mL of Vitox [Oxoid, SR0090A] per liter of agar), and on fastidious anaerobe agar (Neogen, NCM2020A, with addition of 50 mL defibrinated horse blood/L). The broth was incubated under anaerobic conditions. The remaining fastidious anaerobe broth, which contained the tissue, was incubated under anaerobic conditions for 3 days, and was then plated again on three media, as described above. All of the incubations were done at 37°C, with the standard total incubation time for the samples of 7 days being prolonged to 14 days in selected cases. The identification of species was performed through matrix-assisted laser desorption ionization-time of flight (MALDI-TOF), using MALDI Biotyper.

The extraction of DNA from the samples was performed using proteinase K, lysozyme, and lysostaphin. 16S-rDNA PCR was performed for the amplification of the gene, using the P5f (5′-TGCCAGCMG CCGCGGTWAT-3′) and P1067r (5′-ACCATYTCACRACACGAGCT-3′) primers, which resulted in an amplified fragment of approximately 570 bp (23). The amplified fragments were visualized and separated using gel electrophoresis, and Sanger sequencing was used to sequence the amplified gene. The sequence was then compared to previously stored sequences in the Nucleotide BLAST database (at https://blast.ncbi.nlm.nih.gov/Blast.cgi) to determine the bacterial origin. For mixed sequences, the Ripseq mixed Sanger poly software (Pathogenomics, Santa Cruz, CA, USA) was used. In addition to 16S, PCR directed toward *S. aureus nuc* using the primers CTG ATA AAT ATG GAC GTG GCT TAG and GCA ACT TTW GCY AAR CCT TGA C was performed on all of the samples.

**Statistical analysis.** The statistical analyses were performed using IBM SPSS. Fisher's exact test was used for the comparisons of categorical variables. A $P$ value of less than 0.05 was considered to be indicative of a statistically significant result.

## SUPPLEMENTAL MATERIAL

Supplemental material is available online only.
**SUPPLEMENTAL FILE 1**, DOCX file, 0.02 MB.

## ACKNOWLEDGMENTS

We thank Emma Söderdahl for the administrative help, Lena Hyllebusk for the help with the retrieval of microbiological data, as well as Karl Oldberg and Ann-Cathrine Petersson for the important comments on microbiological methods. This research was supported by the Swedish governmental funds for clinical research to M.R., by a postdoctoral grant from the Swedish Society for Medical Research, and by project grants from the Swedish Medical Association, the Thelma Zoégas Foundation for Medical Research, and the Mats Kleberg Foundation to T.S. A part of this work was presented as a part of an abstract at the ISCVID meeting at the University of Barcelona (18th to 20th of June 2022).

We declare no conflicts of interest that are relevant to the content of this article.

G.J. contributed to the design of the study, collected clinical data, performed the analyses, and drafted the manuscript. T.S. contributed to the design and contributed microbiological expertise. S.R. contributed to the design and contributed thoracic surgery expertise. M.R. designed the study, provided tutoring, and revised the manuscript. All authors provided feed-back on the writing of the manuscript and consented to the final version.

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
