## [Reviewer comments · Microbiology Spectrum]

Microbiology Spectrum

Clinical Significance of 16S-rDNA Analysis of Heart Valves in Patients with Infective Endocarditis - a Retrospective Study

Gustav Johansson, Torgny Sunnerhagen, Sigurdur Ragnarsson, and Magnus Rasmussen

Corresponding Author(s): Magnus Rasmussen, Lunds University

Review Timeline:

Submission Date:	March 22, 2023
Editorial Decision:	April 20, 2023
Revision Received:	April 27, 2023
Accepted:	April 28, 2023

Editor: MARK PANDORI

Reviewer(s): Disclosure of reviewer identity is with reference to reviewer comments included in decision letter(s). The following individuals involved in review of your submission have agreed to reveal their identity: Georg Conrads (Reviewer #1); Todd Kitten (Reviewer #2); Yue Zheng (Reviewer #3)

Transaction Report:

DOI: <https://doi.org/10.1128/spectrum.01136-23>

April 20, 2023

Prof. Magnus Rasmussen
Lunds University
Infection medicine
BMC B14, Tornavägen 10
Lund 22184
Sweden

Re: Spectrum01136-23 (Clinical Significance of 16S-rDNA Analysis of Heart Valves in Patients with Infective Endocarditis - a Retrospective Study)

Dear Prof. Magnus Rasmussen:

Your manuscript has been reviewed by three members of the scientific community. Their comments and suggestions for modification are below.

Link Not Available

Sincerely,

MARK PANDORI

Journals Department
Reviewer comments:

Reviewer #1 (Comments for the Author):

This is a very important, nice and well-written study.
To improve it further, I recommend the following changes/revisions:

-Materials and methods should be improved:

Please describe the statistical tests you performed as you report significance levels.

Please describe the method of heart valve sampling (biopsy, instrument, size of biopsy, any details would be good to know).

The type of agar plates is described superficially, what type of blood agar? For instance, chocolate agar is usually required for the HACEK group, but not mentioned here. What is the "fastidious anaerobic agar"; please contact the laboratory for details and edit this chapter.

16S amplification/sequencing: How did you detect a mixed infection (e.g. *S. aureus* together with *S. agalactiae* or *E. faecalis*), as we can usually hardly resolve a mixed 16S sequence?

Results

Viridans streptococci are a frequently used term in the clinical evaluation of a microbiological pathogen, but should be extended here to the - as far as possible - exact species (possibly as appendix material). The same applies to HACEK group, enterococci and β -haemolytic streptococci; please specify.

Discussion.

Please expand the interesting discussion on the contamination risk of valve cultures to include the two findings of *Acinetobacter* spp. and *Pseudomonas oryzihabitans* (I suspect all water contamination; similar as "Environmental bacteria").

Otherwise well done.

Reviewer #2 (Comments for the Author):

The study by Johansson et al investigates the microbiology of infective endocarditis at the authors' institution and the benefit provided by 16S rDNA analysis of surgically removed heart valves. The study will be of considerable interest to some readers. The article is generally well written, but I believe it would benefit from the changes suggested below.

- 1) Line 98: It may be helpful to add "(culture or 16S)" after "valve analysis" so that the reader understands that "one" and "the other" are referring to the different analyses performed on valves rather than to different valves.
- 2) Line 108: If I understand this sentence, the meaning would be clearer if the word "Previously" was replaced with "Prior to that," to indicate that the authors are referring to practices prior to 2019.
- 3) Line 112: The meaning of this statement is unclear. Presumably the valve tissue was homogenized in some fashion. Could the authors elaborate? Also, was there a set procedure for determining the portion of valve tissue allocated to culture vs. PCR?
- 4) Line 120-1: Can the authors provide a reference for these primers? It would be especially helpful if the paper cited characterized primer specificity.
- 5) Line 123: How did the authors determine when lysozyme and lysostaphin were needed? Was this based on failure of the PCR without them, the species (based on culture results), or some other criterion?
- 6) Lines 125-6: Which database was used--a custom database created in-house, a general nucleotide database, or a public database devoted to rRNA sequences?
- 7) Line 140: Although the finding that valve cultures were far more frequently negative than blood cultures or 16S valve analyses is apparently not novel, identification of the cause of this difference would be a valuable contribution. Have the authors examined whether there was a difference in the duration of antibiotic treatment prior to attempted valve culture for the samples that were positive vs. negative? Such a relationship might be apparent for the patient population as a whole or when broken down by organism.
- 8) Line 147 and elsewhere: The results of statistical tests are provided, but the tests employed have not been identified.
- 9) Lines 252-4: I don't understand why 16S analysis not being performed on 3 patients with BCNE indicates a less severe bias than originally thought. Are the authors suggesting that this number is smaller than they had expected, larger, or is there some other reasoning?

Reviewer #3 (Comments for the Author):

The authors claimed that 16S-analysis should be routinely performed on surgically removed valves in blood culture-negative endocarditis. In patients with positive blood cultures, 16S analysis may also be considered since a diagnostic benefit was provided in some patients. However, I have some concerns.

1. What are immune cells (eg. macrophages) abundance and response in patients with culture-negative and positive endocarditis?
2. Please do a small prospective analysis (ie. 10 culture-negative: 10 positive endocarditis patients) to validate your retrospective study findings.
3. if possible, please collect the patients' PBMC and heart surgery samples to do RNA-seq to investigate the immune responses and mechanisms, which may be potential targets to treat the patients.

Staff Comments:

Preparing Revision Guidelines

Please return the manuscript within 60 days; if you cannot complete the modification within this time period, please contact me. If you do not wish to modify the manuscript and prefer to submit it to another journal, please notify me of your decision immediately so that the manuscript may be formally withdrawn from consideration by Microbiology Spectrum.

Dear Dr Pandori,

We thank you for your letter and for the constructive criticism brought forward by the reviewers. We have modified our manuscript according to their suggestions and we feel that this has improved the quality of the manuscript. The detailed response is given below.

Best regards

Magnus Rasmussen

Reviewer #1: This is a very important, nice and well-written study. To improve it further, I recommend the following changes/revisions:

-Materials and methods should be improved:

Please describe the statistical tests you performed as you report significance levels.

Author reply: Thank you for this comment. We have added a section on statistics to the methods section and we have also indicated, in the result section, which test that was used for specific comparisons throughout the manuscript.

Reviewer #1: Please describe the method of heart valve sampling (biopsy, instrument, size of biopsy, any details would be good to know).

Author reply: Details about the procedure have been added to the method section.

Reviewer #1: The type of agar plates is described superficially, what type of blood agar? For instance, chocolate agar is usually required for the HACEK group, but not mentioned here. What is the "fastidious anaerobic agar"; please contact the laboratory for details and edit this chapter.

Author reply: We thank the reviewer for this chance to specify which plates are used. We have made appropriate specifications in the text.

Reviewer #1: 16S amplification/sequencing: How did you detect a mixed infection (e.g. *S. aureus* together with *S. agalactiae* or *E. faecalis*), as we can usually hardly resolve a mixed 16S sequence?

Author reply: Thank you for this important comment. We have clarified this in the methods section.

Reviewer #1: Results

Viridans streptococci are a frequently used term in the clinical evaluation of a microbiological pathogen, but should be extended here to the - as far as possible - exact species (possibly as appendix material). The same applies to HACEK group, enterococci and β -haemolytic streptococci; please specify.

Author reply: We are grateful for this comment and we completely agree with the suggestion. We have added the species and group determination of all bacteria as a new Appendix 3.

Reviewer #1: Discussion.

Please expand the interesting discussion on the contamination risk of valve cultures to include the two findings of *Acinetobacter* spp. and *Pseudomonas oryzihabitans* (I suspect all water contamination; similar as "Environmental bacteria").

Otherwise well done.

Author reply: We have added a sentence on this matter to the discussion.

Reviewer #2 (Comments for the Author):

The study by Johansson et al investigates the microbiology of infective endocarditis at the authors' institution and the benefit provided by 16S rDNA analysis of surgically removed heart valves. The study will be of considerable interest to some readers. The article is generally well written, but I believe it would benefit from the changes suggested below.

1) Line 98: It may be helpful to add "(culture or 16S)" after "valve analysis" so that the reader understands that "one" and "the other" are referring to the different analyses performed on valves rather than to different valves

Author reply: Thank you for this good suggestion. This has been added.

Reviewer 2: 2) Line 108: If I understand this sentence, the meaning would be clearer if the word "Previously" was replaced with "Prior to that," to indicate that the authors are referring to practices prior to 2019.

Author reply: We have made the changes suggested.

Reviewer 2: 3) Line 112: The meaning of this statement is unclear. Presumably the valve tissue was homogenized in some fashion. Could the authors elaborate? Also, was there a set procedure for determining the portion of valve tissue allocated to culture vs. PCR?

Author reply: We have added details on how this procedure was performed.

Reviewer 2: 4) Line 120-1: Can the authors provide a reference for these primers? It would be especially helpful if the paper cited characterized primer specificity.

Author reply: A reference for the primers has been provided.

Reviewer 2: 5) Line 123: How did the authors determine when lysozyme and lysostaphin were needed? Was this based on failure of the PCR without them, the species (based on culture results), or some other criterion?

Author reply: The question is very relevant and we have made changes to the text to correctly reflect the procedure.

Reviewer 2: 6) Lines 125-6: Which database was used--a custom database created in-house, a general nucleotide database, or a public database devoted to rRNA sequences?

Author reply: Changes have been made to address this relevant question.

Reviewer 2: 7) Line 140: Although the finding that valve cultures were far more frequently negative than blood cultures or 16S valve analyses is apparently not novel, identification of the cause of this difference would be a valuable contribution. Have the authors examined whether there was a difference in the duration of antibiotic treatment prior to attempted valve culture for the samples that were positive vs. negative? Such a relationship might be apparent for the patient population as a whole or when broken down by organism.

Author reply: This is a relevant comment but we feel that this would put the focus on the culture method and the purpose of this study was to investigate the usefulness of the 16S-analysis, not the cultures. No changes have been made.

Reviewer 2: 8) Line 147 and elsewhere: The results of statistical tests are provided, but the tests employed have not been identified.

Author reply: This is our mistake. Please also see reply to reviewer 1. Relevant changes have been made to the manuscript.

Reviewer 2: 9) Lines 252-4: I don't understand why 16S analysis not being performed on 3 patients with BCNE indicates a less severe bias than originally thought. Are the authors suggesting that this number is smaller than they had expected, larger, or is there some other reasoning?

Author reply: We have deleted this speculation. The reasoning was that cases of BCNE would be more likely subjected 16S-analysis than cases of blood culture positive IE before 2019. The finding of three cases of BCNE among cases not subjected to 16S would indicate that it was not necessarily so that 16S-analysis was performed on all cases of BCNE.

Reviewer #3 (Comments for the Author):

The authors claimed that 16S-analysis should be routinely performed on surgically removed valves in blood culture-negative endocarditis. In patients with positive blood cultures, 16S analysis may also be considered since a diagnostic benefit was provided in some patients. However, I have some concerns.

1. What are immune cells (eg. macrophages) abundance and response in patients with culture-negative and positive endocarditis?

Author reply: It is unclear to us why this is a concern of the reviewer. The question appears to be related to a completely different study. No changes have been made in relation to this comment.

Reviewer 3: 2. Please do a small prospective analysis (ie. 10 culture-negative: 10 positive endocarditis patients) to validate your retrospective study findings.

Author reply: We cannot see what difference this would make for our conclusions or results. No changes have been made.

3. if possible, please collect the patients' PBMC and heart surgery samples to do RNA-seq to investigate the immune responses and mechanisms, which may be potential targets to treat the patients.

Author reply: This comment refers to a completely different study as compared to the one that we have conducted. No changes have been made.

April 28, 2023

Prof. Magnus Rasmussen
Lunds University
Infection medicine
BMC B14, Tornavägen 10
Lund 22184
Sweden

Re: Spectrum01136-23R1 (Clinical Significance of 16S-rDNA Analysis of Heart Valves in Patients with Infective Endocarditis - a Retrospective Study)

Dear Prof. Magnus Rasmussen:

I have assessed the Reviewers' comments, and the subsequent author responses to the comments. The relevant issues have been addressed.

Your manuscript has been accepted, and I am forwarding it to the ASM Journals Department for publication. You will be notified when your proofs are ready to be viewed.

Sincerely,

MARK PANDORI
Editor, Microbiology Spectrum
